# Research on Improved Quantitative Identification Algorithm in Odor Source Searching Based on Gas Sensor Array

**DOI:** 10.3390/mi14061215

**Published:** 2023-06-08

**Authors:** Yanru Zhao, Dongsheng Wang, Xiaojie Huang

**Affiliations:** The College of Mechanical and Power Engineering, Henan Polytechnic University, Jiaozuo 454003, China; wangdongsheng@hpu.edu.cn (D.W.); 211705010024@home.hpu.edu.cn (X.H.)

**Keywords:** gas sensor array, neural network, quantitative identification, odor source searching

## Abstract

In order to improve the precision of gas detection and develop valid search strategies, the improved quantitative identification algorithm in odor source searching was researched based on the gas sensor array. The gas sensor array was devised corresponding to the artificial olfactory system, and the one-to-one response mode to the measured gas was set up with its inherent cross-sensitive properties. The quantitative identification algorithms were researched, and the improved Back Propagation algorithm was proposed combining cuckoo algorithm and simulated annealing algorithm. The test results prove that using the improved algorithm to obtain the optimal solution −1 at the 424th iteration of the Schaffer function with 0% error. The gas detection system designed with MATLAB was used to obtain the detected gas concentration information, then the concentration change curve may be achieved. The results show that the gas sensor array can detect the concentration of alcohol and methane in the corresponding concentration detection range and show a good detection performance. The test plan was designed, and the test platform in a simulated environment in the laboratory was found. The concentration prediction of experimental data selected randomly was made by the neural network, and the evaluation indices were defined. The search algorithm and strategy were developed, and the experimental verification was carried out. It is testified that the zigzag searching stage with an initial angle of 45° is with fewer steps, faster searching speed, and a more exact position to discover the highest concentration point.

## 1. Introduction

In today’s era of rapid development of various technologies, the quantitative detection and analysis of gas have broad application prospects in the fields of food science, environmental science, public security, and national defense and military [1,2,3]. Due to the widespread disadvantages of cross sensitivity and selectivity difference, it is difficult to realize the detection and analysis of mixed gases with a single sensor [4]. At present, more methods have been used to form a sensor array through multiple gas sensors with different sensitivity degrees and combine with the mode recognition theory and method for gas analysis [5,6,7]. Current work focuses on the qualitative detection of multi-component gas, while quantitative analysis is less involved [8].

The bionic smell is sampled based on the process of smell obtained by natural organisms, which converts various odors in life into digital information through the comprehensive use of a sensor array, signal processing system, and computer identification technology [9]. With the development of life science and computer science, the research of bionic robots has received more and more attention. Its main research is to imitate various biological perception systems and design robot navigation systems based on biological vision, hearing, and smell mechanisms [10,11]. At present, there are many studies of visual and auditory aspects, while the study of olfactory robots imitating bionic smell, namely, active smell, is just starting [12]. Compared with the traditional passive application of the artificial olfactory system that can only sense the odor, the data detected by the sensors also can be integrated to track and locate the odor sources [13].

Scholars at home and abroad have carried out a lot of research work on mobile robot odor source searching. However, the work focus is mostly on the selection and formulation of search strategy, multi-sensor cooperation mode, saving search time, and so on [14,15,16]. Not much is involved in how to improve the accuracy of gas concentration detection, so the study starts from the perspective of improved quantitative gas concentration identification algorithms and combines the relevant search strategy to improve the search accuracy.

To improve the accuracy of gas concentration detection, the improved BP algorithm based on the combination of the cuckoo algorithm and simulated annealing algorithm was proposed. The improved algorithm reduces the dependence of the BP network on the initial weight and threshold and overcomes the disadvantage of falling into the local optimal solution. The search algorithm was designed, and the search strategy was formulated. The experimental scheme was designed, and the experimental platform was built. The sensor voltage and gas concentration information were normalized to meet the input and output requirements of the neural network. The mobile robot odor source search scheme based on the gas sensor array uses fewer steps, faster search speed, and more accuracy to find the location of the maximum concentration point.

## 2. Gas Sensor Array Construction and Neural Network Design

### 2.1. Gas Sensor Array Construction and Calibration

The gas sensor array equivalent to primary olfactory neurons is composed of multiple independent gas sensors. Due to the inherent cross-sensitive characteristics of the gas sensor, the response of the array can establish a one-to-one response pattern to the gas in the current environment [17]. Six independent metal oxide semiconductor gas sensors were selected to form a symmetrical sensor array on the left and right sides, respectively, TGS2611 and TGS2620 of Figaro Company of Japan and MP-4 sensors of Zhengzhou Weisheng Technology Co., Ltd. of China. Both TGS series and MP-4 sensors have good sensitivity to the measured gas in a relatively wide range of concentrations, with cross-sensitive to different categories of gas, such as alcohol and methane, respectively. The selected sensors adopt side heat heating. The heating loop and test loop are independent, which avoids voltage interference and can obtain relatively accurate test results. Applying an appropriate voltage to the heating can keep the sensor just at the operating temperature. The performance parameters of these three classes of sensors are shown in Table 1.

The gas sensor array detection module combined with the neural network module can realize the quantitative detection of the gas concentration in the environment only by obtaining the output voltage of the gas sensor array. The overall schematic diagram of the gas sensor array detection module is shown in Figure 1.

According to the experimental environment and controllability requirements, the four-wheel mobile platform used in the field of mobile cars was selected as the mechanical carrier of the gas sensor array detection module. Through the analysis of the artificial olfactory system and the research of the olfactory robot, the designed olfactory robot includes the following modules, sensor array, filter circuit, pattern recognition system, obstacle avoidance circuit module, etc. In this paper, only the quantitative identification of the gas concentration in the environment and the strategy of searching for the odor source were studied. The gas sensor array detection module and the designed olfactory robot are shown in Figure 2.

The calibration of the sensor refers to a process of establishing the relationship between the input and output of the sensor and determining the error under different use conditions. The output voltage of the sensors and gas concentration are used to train the neural network to establish a non-linear input and output relationship to realize the calibration of the sensor array. Therefore, to achieve the calibration of the sensor array, it is necessary to select an appropriate neural network, then design the neural network according to the input and output of the sensor array to realize the subsequent quantitative identification of the gas concentration.

### 2.2. Improved Quantitative Identification Algorithm Propose and Design

Feedforward neural network is an artificial neural network based on tutor learning, which can acquire experience from the learning of known samples and thus have the ability to identify unknown samples. BP (Back Propagation) neural network is a more commonly used multi-layer feedforward neural network, which is widely used in function approximation, pattern recognition, and other fields. Meanwhile, it has good data fitting ability and high modeling ability of complex nonlinear systems. The classical BP algorithm is essentially a simple static optimization algorithm for the fastest descent, which has the disadvantages, such as easily falling into the local optimal value and slow convergence speed. In addition, the merits and demerits of the BP algorithm also have a great relationship with the initial weights and threshold.

For the disadvantages of the classical BP algorithm easily falling into local optimum and relying on initial weight and threshold, the improved BP algorithm is proposed for the quantitative identification of gas concentration in the environment. The improved BP algorithm uses the combination of the cuckoo algorithm and the simulated annealing algorithm to obtain the optimal cuckoo nest and takes this as the initial weight and threshold. After repeated training until the minimum prediction error is obtained to obtain the optimal initial weight and threshold of the neural network.

In essence, the simulated annealing algorithm belongs to the search algorithm of the Markov chain under appropriate conditions. In the process of algorithm iteration, the common Metropolis criterion is adopted to continuously produce new solutions, judge, accept or abandon operations and can accept a solution worse than the current solution with a certain probability until the optimal solution is obtained [18]. A schematic diagram of the algorithm is shown in Figure 3.

The cuckoo algorithm combining stochastic algorithm and local search algorithm is a natural heuristic algorithm proposed by Xin-She Yang and Suash Deb in 2009. It is a random search algorithm that simulates the biological behavior of cuckoo parasitic hatching chicks in nature, which reflects the law of natural selection and survival of the fittest in nature. The algorithm only has two key parameters, the probability and population of exotic eggs being found, which are simple and easy to implement. The cuckoo algorithm effectively combines the difference algorithm, the particle swarm algorithm and the simulated annealing algorithm with global convergence. The cuckoo algorithm uses a balanced combination of local random walks controlled by switching parameters and a globally explored random walk. The local random walks are as follows:(1)$$x_{i}^{t+1}=x_{i}^{t}+\alpha s⊗H(p_{a}-\varepsilon )⊗(x_{j}^{t}-x_{k}^{t})$$

$x_{j}^{t}$ and are the two different solutions selected by random permutation, *H*(*u*) is the step function, *ε* is the random number drawn from a uniform distribution, *s* is the step size, and *α* is the scaling factor of step size.

The global random walk uses a Levy flight with strong randomness, namely:(2)$$x_{i}^{t+1}=x_{i}^{t}+\alpha ⊗levy(\lambda )$$
where $x_{i}^{t}$ is the position of the *i*-th nest in the *t*-generation and the *levy*(*λ*) is the random step size obeying the Levy distribution.

The random steps are generated with the *Mantegna* algorithm, where the step *s* is as follows:(3)$$s=\frac{u}{{\left|v\right|}_{}^{1/\beta }}$$
(4)$$u\sim N(0,\sigma_{u}^{2}),v\sim N(0,\sigma_{v}^{2})$$
(5)$$\sigma_{}^{u}={\left\{\frac{\Gamma (1+\beta )\sin(\pi \beta /2)}{\Gamma [(1+\beta )/2]\beta_{}^{2(\beta -1)/2}}\right\}}_{}^{1/\beta }$$

In most cases, *α* = *O*(*L*/10); *L* is the characteristic scale of the problem of interest.

The intelligent algorithm basic test function *Schaffer* was compiled in MATLAB (v2013). The advantages and disadvantages of the improved algorithm were verified and contrasted by finding the optimal solution to test the cuckoo algorithm (CS), cuckoo algorithm introducing the simulated annealing algorithm (CS-SA), and the improved algorithm combining the cuckoo algorithm and simulated annealing algorithm by introducing an adaptive adjustment strategy for discovery probabilities.

The total number of iterations of the algorithms was set to 1000 times, and the independent variable range of the test function was set to [–10, 10]. The first two algorithms take the functional values of the test function as the fitness values of the cuckoo algorithm. The improved algorithm takes new fitness values and selection probabilities generated from the currently obtained maximum and minimum fitness values by introducing memory functions as a measure of determining whether a target nest is discarded. The optimal fitness value obtained finally after the iteration is the target value of the function. The position of the optimal nest obtained is the horizontal and vertical coordinate value corresponding to the minimum value of the objective function.

The simulation results of the three different algorithms for finding the minimum value of the test function are shown in Figure 4.

The results obtained from Figure 4 are as follows. Using only the cuckoo algorithm, no optimal solution is obtained at the 1000th iteration, and the optimal solution obtained at the end of the iteration is −0.99996 with an error of 0.004%. After introducing the simulated annealing algorithm, the optimal solution at the 527th iteration is −1 with an error of 0%. After introducing an adaptive adjustment strategy for the solution to the cuckoo algorithm, the optimal solution at the 424th iteration is −1 with an error of 0%. The three algorithms obtaining the convergence rate of the optimal solution of the function are accelerated successively. The improved algorithm is with high convergence speed and small error, which achieves the expected improved effect.

### 2.3. Neural Network Design

In the experiment, a mixture of saturated steam of alcohol and pure methane was used. A 5 L sealed container made of an acrylic plate with 0.5 mm thickness was taken as a measuring container. The one-by-one response mode formed by the output voltage of the sensor array and the concentration of the gas to be measured can be taken as the actual input and the target output of the neural network after the signal preprocessing [19]. The gas concentrations selected for neural network training and prediction range from 0 to 2000 ppm. The final selected experimental data totaled 11 × 11 = 121 groups.

In order to realize the quantitative identification of gas concentration, six gas sensors are used to form two symmetrical sensor arrays, for which two identical neural networks should be designed for the detection of gas concentration. The training function selects the *trainscg* function, which can avoid time-consuming 1D search and accelerate the convergence speed of the network [20]. The training target value of the two neural networks is set to 0.0001. That is, when the number of neural network training times reaches the maximum value of 1000 or the error sum of squares of the neural network drops below 0.0001, the training is terminated. After the network training is completed, 23 groups of experimental data randomly selected are input into the two neural networks for prediction. The average value of the mean squared error of the two gases is used as the fitness to determine whether the optimal weight and threshold are found. The relative error is compared as an evaluation criterion for neural networks to contrast the predictive results. It is shown that the two neural networks achieve the convergence target after 1121 and 1214 iterations, respectively. The predictive results of the quantitative recognition by one of the neural networks are shown in Table 2.

According to Table 2, the predictive curve of alcohol steam of the first neural network is shown in Figure 5a, and the result for methane is in Figure 5b. The error between the predictive gas concentration and the expected (actual) concentration can be intuitively seen from the two predictive curves. The prediction error obtained with the improved algorithm is significantly smaller than that of the BP algorithm, and the BP algorithm optimized only with the cuckoo algorithm; thus, the advantages of the improved algorithm are proved.

It is found from Table 2 that the maximum predictive error of the neural network for alcohol steam is 24.731%, the average predictive error is 8.225%, and those for methane are 22.218% and 4.911%, respectively. Although the maximum predictive error is large, it can be seen from Figure 5 that in the concentration curve predicted by the improved algorithm, the number of large error points is a small proportion of the total prediction number. The points for the relative error larger than 15% of the predictive concentration of two different gases are both 2 and account for 8.696% of the total prediction number. On the whole, the gas sensor array shows fine detection performance when detecting the concentration of the above two gases within the corresponding concentration detection range.

The structure of the BP neural network completed is shown in Figure 6. Where r_1_, r_2_, and r_3_ represent the output voltage of the sensor array after normalization. Y_1_ and y_2_ indicate the gas concentration acquired.

## 3. Odor Source Searching Algorithm Design and Experiment Verification

In the study, the active olfactory method is applied, that is, a mobile robot carrying a sensor module to search for odor sources. The Gaussian model is taking as the gas propagation model, and the concentration distribution formula of the Gaussian plume model is as follows:(6)$$C(x,y,z,H)=\frac{Q}{2\pi \sigma_{y}\sigma_{z}\bar{u}}\exp(-\frac{y^{2}}{\sigma_{y}^{2}})\left\{\exp\left[-\frac{{\left(Z-H\right)}^{2}}{2\sigma_{z}^{2}}\right]+\exp\left[-\frac{{\left(Z+H\right)}^{2}}{2\sigma_{z}^{2}}\right]\right\}$$
where *C* is the pollutant concentration at any point; *H* is the effective height of the discharge outlet; *Q* is the source strength, that is, the pollutant discharge per unit of time; $\bar{u}$ is the average wind speed; and *σ_y_* and *σ_z_* are horizontal and vertical diffusion coefficients, respectively.

The key to calculate the pollutant concentration with Formula (6) is to accurately determine the values of *σ_y_* and *σ_z_*, which are related to the stability of the atmosphere. At a given level of stability, the diffusion coefficients are functions of the distance *x* to the center of the chimney, namely:(7)$$\sigma_{y}=ax^{b},\sigma_{z}=cx^{d}$$

The diffusion coefficients with the atmospheric stability level F are selected, that is, *a* = 0.055, *b* = 0.929, *c* = 0.062, and *d* = 0.784. The average wind speed is set to 0.5 m/s, and the volume of the simulated environment is 6 m × 8 m × 4 m. The leakage source is the elevated point source diffusion with a certain distance from the ground level; the height is selected as *H* = 0.5 m. The source strength of alcohol steam is set to 50 ppm/s and converted to the mass flow *Q* = 102.835 mg/s. The diffusion concentration chart of the Gaussian plume model is plotted with MATLAB according to the parameter settings, and the ground concentration distribution at *Z* = 0 is shown in Figure 7.

It is demonstrated from Figure 7 that the gas leakage source is the point source diffusion, and the concentration in the *x* direction is normally distributed on the *y* axis. On the *x* axis, the gas concentration is close to zero near the pollution source at the ground. Along the downwind direction, the concentration on the ground axis reaches the maximum at a certain distance from the point source and gradually decreases.

By solving the derivative and taking the extreme for Formula (8), the maximum concentration *C*_max_ and the distance *x*_max_ from the leakage source can be obtained. When *y* = 0, *z* = 0, the concentration distribution along the *x*-axis can be achieved.
(8)$$C(x,0,0,H)=\frac{q}{\pi \bar{u}\sigma_{y}\sigma_{z}}\exp\left(-\frac{H^{2}}{2\sigma_{z}^{2}}\right)$$

Make ∂*C*/∂*x* = 0, since both σ*_y_*and σ*_z_* are unknown functions of *x*, assume σ*_y_*/σ*_z_*= *m* (*m* is a constant), namely:(9)$$\frac{\partial }{\partial \sigma_{z}}\left(\frac{q}{\pi \bar{u}m\sigma_{z}^{2}}\exp\left(-\frac{H^{2}}{2\sigma_{z}^{2}}\right)\right)=\exp\left(-\frac{H^{2}}{2{\left(m\sigma_{z}\right)}^{2}}\right)\frac{q}{\pi \bar{u}m\sigma_{z}^{3}}\left(-2-\frac{H^{2}}{{\left(m\sigma_{z}\right)}^{2}}\right)=0$$

When,
(10)$$\sigma_{Z}{|}_{x=x_{\max}}=H/\sqrt{2}$$

The maximum concentration may be computed below.
(11)$$C_{\max}=\frac{2q}{\pi e\bar{u}H^{2}}=\frac{\sigma_{z}}{\sigma_{y}}$$

During the simulation experiment, the initial step size of the robot is set to *d*_0_ = 0.8 m. To simplify the calculation, the termination search condition defined in the experiment is that the gas concentration detected in step *n* is less different from the maximum concentration in the plume model. The difference in horizontal-vertical coordinates between the final point and the maximum concentration point should not be greater than 0.08 m. The rules of mobile robots searching for odor sources always move towards a large rate of concentration change.

Figure 8 shows the change of the detected gas concentration with the search steps during the odor source searching with different initial steering angles. It can be seen that the gas concentration detected by mobile robots during the search process generally shows an increasing trend, and the step length and steering angle can be timely adjusted according to the concentration change. During the first phase of the Z-shaped searching with the initial steering angle 45°, the maximum concentration point is found at step 16. The maximum concentration is 293.229 mg/m^3^, and the error is 0%. When searching at a 30° initial steering angle, the maximum concentration point is found at step 19, and the maximum concentration is 293.101 mg/m^3^; the error is 0.044%. When searching with a 60° initial steering angle, the maximum concentration point is found at step 19. The maximum concentration is 293.229 mg/m^3^; the error is 0%. When using a 75° steering angle, the maximum concentration point is found at step 24; the maximum value is 293.229 mg/m^3^, and the error is 0%. In the first stage, four different initial angles are adopted to find that the error of the maximum concentration is within the allowable range so the design requirements are satisfied.

## 4. Conclusions

The gas sensor array was designed based on the artificial olfactory system, using the inherent cross-sensitivity characteristics to form a high-dimensional response to the measured gas to provide sufficient information to express the gas environment. An improved algorithm was proposed for the quantitative identification of gas. It is proved that the optimal solution −1 of the *Schaffer* function was obtained in the 424th iteration by the improved algorithm, and the error is 0%. The experimental scheme was designed, and the experimental platform was built to analyze and compare the predictive results of the two groups of neural networks. The Gaussian plume model was selected as the plume diffusion model, and the search algorithm was designed with MATLAB. It is shown that when the initial angle of 45° was used for Z-shaped searching, the maximum concentration point was found at step 16, the steps were fewer, the searching speed was faster, and the location of the maximum concentration point was more accurate.

## Figures and Tables

**Figure 1 micromachines-14-01215-f001:**
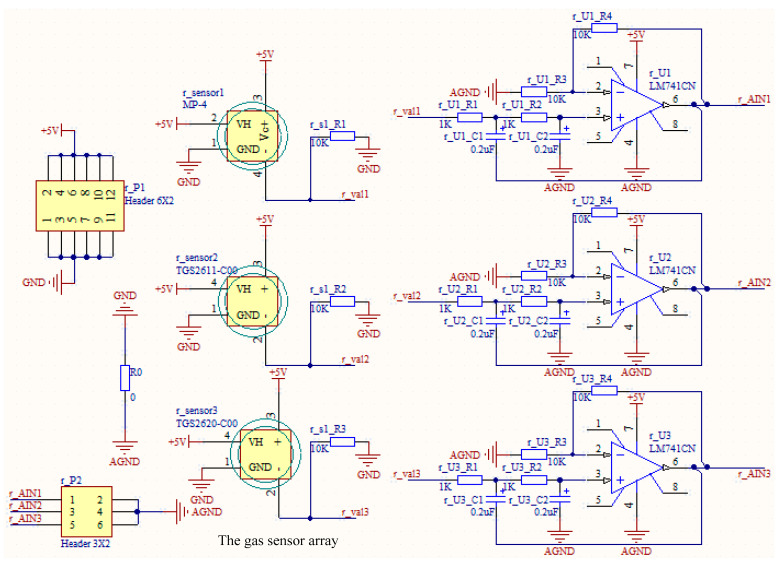
The schematic diagram of the gas sensor array detection module.

**Figure 2 micromachines-14-01215-f002:**
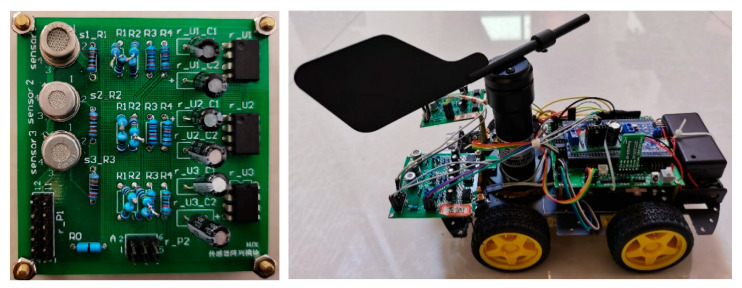
The gas sensor array detection module and the designed olfactory robot.

**Figure 3 micromachines-14-01215-f003:**
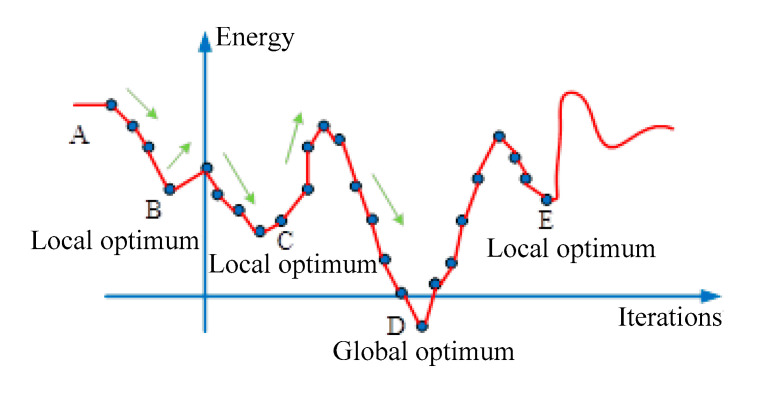
Schematic diagram of simulated annealing algorithm.

**Figure 4 micromachines-14-01215-f004:**
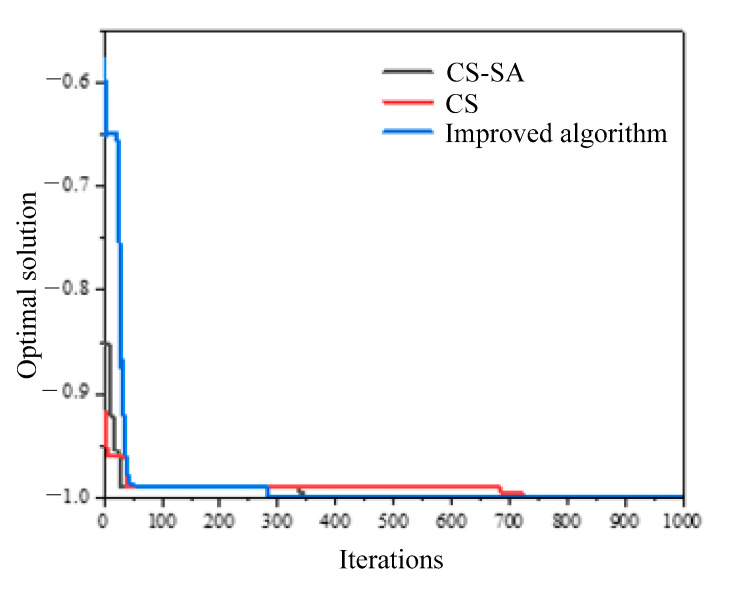
Results comparison of *Schaffer* test function for different algorithms.

**Figure 5 micromachines-14-01215-f005:**
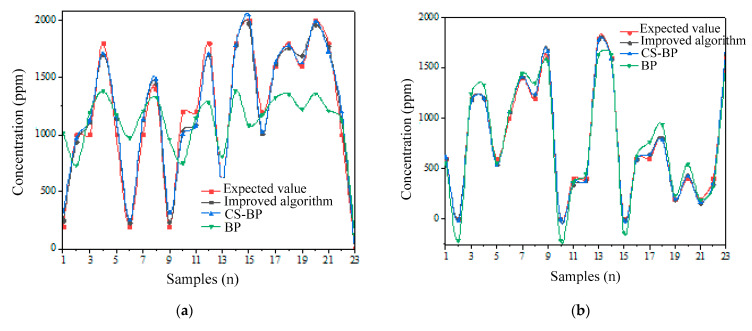
Comparison of predictive curve of alcohol vapor and methane concentration by different algorithms. (**a**) Alcohol vapor; (**b**) CH_4_.

**Figure 6 micromachines-14-01215-f006:**
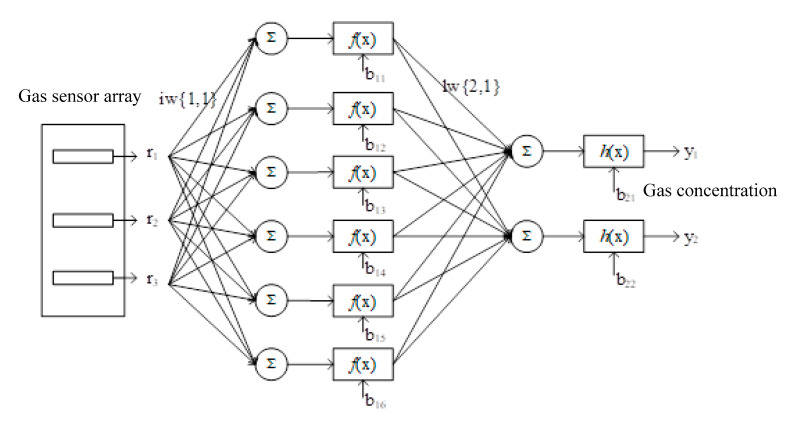
Scheme of neural network.

**Figure 7 micromachines-14-01215-f007:**
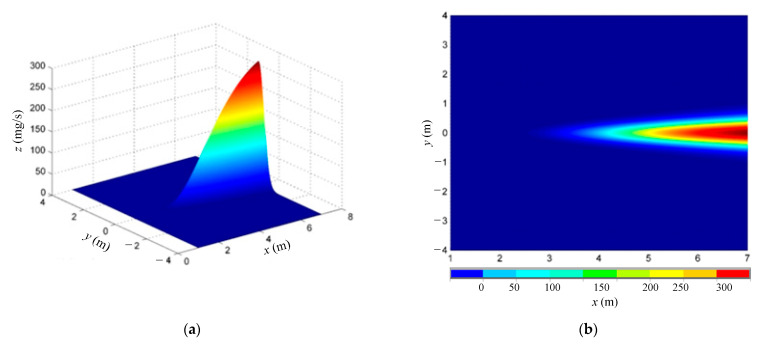
Distribution of plume diffusion concentration. (**a**) 3D image; (**b**) *x*o*y* plane projection.

**Figure 8 micromachines-14-01215-f008:**
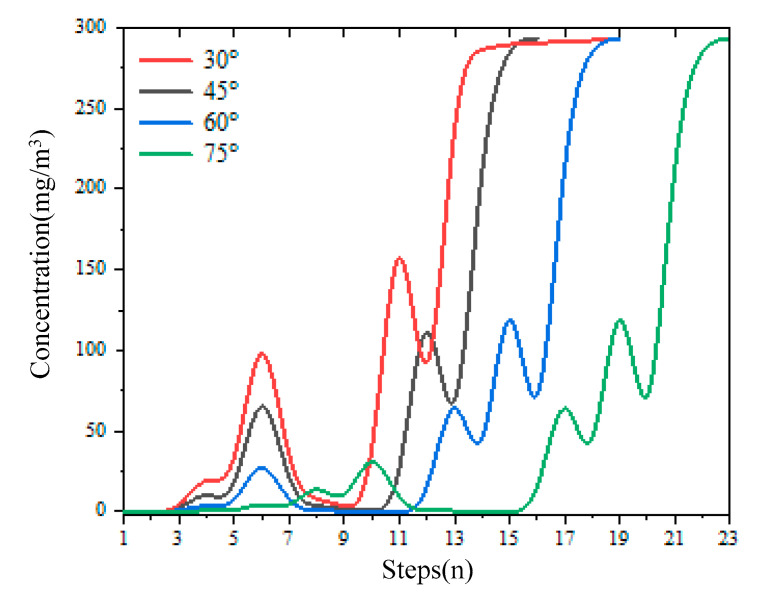
The variation curve of concentration.

**Table 1 micromachines-14-01215-t001:** The performance parameters of three sensors.

Sensor Model	Sensitive Gas	Detection Concentration /ppm	Main Application Areas
TGS2611	Methane, Natural gas	500–10,000	Portable gas detector, Gas leakage detection
TGS2620	Alcohol	50–5000	Ethanol detector
MP-4	Methane, Natural gas, Biogas	300–10,000	Fire prevention, Safety detection system, Gas leak detector

**Table 2 micromachines-14-01215-t002:** Predictive results of the quantitative recognition.

Sample Number	Actual Concentration /ppm		Predictive Concentration /ppm		Relative Error	
	Alcohol	Methane	Alcohol	Methane	Alcohol	Methane
15	200	600	249.461	601.364	0.247	0.002
56	1000	0	938.766	7.656	0.061	—
62	1000	1200	1114.090	1186.792	0.114	0.011
106	1800	1200	1699.340	1202.177	0.056	0.002
59	1000	600	1142.322	541.263	0.142	0.098
17	200	1000	229.694	1066.200	0.148	0.066
63	1000	1400	1126.808	1412.638	0.127	0.009
84	1400	1200	1450.702	1236.539	0.036	0.030
20	200	1600	230.013	1675.077	0.150	0.047
67	1200	0	1030.963	−3.478	0.141	—
69	1200	400	1085.969	336.564	0.095	0.159
102	1800	400	1694.531	380.455	0.059	0.049
43	600	1800	591.252	1788.003	0.015	0.007
108	1800	1600	1783.638	1589.684	0.009	0.006
111	2000	0	1972.889	−6.885	0.014	—
70	1200	600	1027.848	580.465	0.143	0.033
92	1600	600	1620.111	643.626	0.013	0.073
103	1800	800	1756.625	799.729	0.024	0.0003
90	1600	200	1694.422	186.427	0.059	0.068
113	2000	400	1958.524	438.380	0.021	0.096
101	1800	200	1773.025	155.565	0.015	0.222
58	1000	400	1120.502	347.112	0.121	0.1322
9	0	1600	−18.593	1631.091	—	0.019

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
