# Peer review of "Research on Improved Quantitative Identification Algorithm in Odor Source Searching Based on Gas Sensor Array"

_micromachines, 2023, doi:10.3390/mi14061215_

Round 1

Reviewer 1 Report

In this work, the authors present a comprehensive overview of a research study aimed at improving the precision of gas detection and developing effective search strategies for odor source detection. The use of a gas sensor array designed to mimic the artificial olfactory system, and the development of an optimization algorithm combining cuckoo search and simulated annealing algorithms, are particularly noteworthy. The methodology employed, which included the use of MATLAB and neural networks, as well as the test plan and evaluation indices, were well-structured and systematic. The findings of the study, particularly the zigzag searching stage with an initial angle of 45º, appear to have practical implications for enhancing gas detection accuracy and efficiency. Overall, this abstract highlights a well-executed study with promising results for the field of gas detection and odor source search.

However, additional details and justifications are needed to support the authors’ claims. I recommend the publication of the manuscript after the following major revisions are addressed.

1. The authors claim to solve the shortcomings in BP algorithm. However, the definition of 'BP' is not clear and this abbreviation also appears in the abstract, which should be clarified.

2. The choice of three sensors seems not well justified. Please provide more information about the selected sensors.

3. I understand that the authors focus on the implementation of an algorithm in this work. However, it seems that the experimental setup is not well presented and explained in detail. I believe this is also a critical part to support the author's claims.

4. Other minor issues. English writing including grammatical mistakes and tense consistency issues should be solved before publication. The z-axis of fig5a is not labeled. Fig 5b should also provide a color bar.

The overall quality of English can be improved, including fixing grammatical mistakes, and the use of consistent tense in the writing, etc.

e.g., "The optimal fitness value (is) finally obtained after the iteration is the target value of the function."

"To solve the shortcomings of the classical BP algorithm, such as easily falling into the local optimal value and relying on the initial weight and the threshold, the improved BP algorithm was proposed for the quantitative identification of the gas concentration in the environment. The improved BP algorithm uses the combination of cuckoo algorithm and simulated annealing algorithm to obtain the optimal nest of cuckoo, and takes that as the initial weight and threshold."

Reviewer 2 Report

The paper describes the development of an algorithm based on a gas sensor array for the detection of gas concentration (alcohol and methane) and the simulation of this algorithm for odor source searching with mobile robots.

First of all, I would like to point out that it is quite difficult to read the paper due to overall low quality of the English language (please see separate report).

Please find here below my comments to the paper.

Abstract & Title

- The title of the paper focuses on odor source searching, but little of the paper is dedicated to this topic (only the latest section), so I would revise it a little

- BP acronym is not defined but it is used in the abstract

- No result on gas concentration quantification is reported in the abstract

- It is not stated that the odor source searching was carried out in a simulated environment

Introduction

- I find the introduction of the paper quite short, even if it provides good explanation of the context. I would mention the fact that gas detection is useful also in the context of disease diagnosis through exhaled breath analysis (lung cancer, asthma, ...)

- I would add or extend the current paragraph by providing a more extensive information about the state of the art and the results obtained by other research groups

- At the end of the introduction, please add a paragraph explaining how you carried out the experiments and how the paper is organized, in order to provide the reader with an overview of the sections present in the paper

 Gas sensor array construction and neural network design

- Can you show me a schematic of the electrical circuit for gas sensors control and measurement?

- Did you use a commercially available device or did you design your own custom PCB? 

- Did you apply any kind of temperature modulation to the sensor?

- What is the meaning of the sentence "write intelligent algorithm basic test function Schaffer in MATLAB"? on page 3?

- How you carried out the simulation results on the three different algorithms?

- Caption of Figure 2 should be on the previous page

- At the beginning of section 2.3 you mention that your container had a "certain" thickness: can you quantify this thickness? 

- It is not clear to me why you mention that you built two symmetrical sensor arrays (3 sensors each), and why you developed two identical neural networks. A single network (as shown in Figure 4) can predict the concentration of both gases, so why is there the need of having two of them?

- What is the training target value (0.0001) that you mention?

- Did you collect data in a different session (different day, different operator) to be used as a test set? At the moment, your training set is composed of data points that are part of the training curve, but I would really appreciate seeing the results on a completely new test set, to better understand how the NN behaves

- how did you prevent overfitting from occurring during the training of the network?

- I would remove unnecessary decimal digits from all the paper (especially from Table 1 and from the predicted concentration and computed errors)

- the description of the neural network parameters and connections is quite redundant, and I would just leave Figure 4

- You mentioned that you trained two different neural networks, but you reported the results only from one of them. Did you carry out any comparison between the networks?

 Odor source searching algorithm and experiment verification

- Based on the title of the paper, this section should be the main focus of the paper, but in reality a small portion of the paper is dedicated to odor source searching

- Odor source searching was performed in a simulation environment. Which challenges do you see in translating the results from a simulation to a real setting?

- Due to the fact that this part was carried out only in a simulated environment, how did you integrate the properties of the gas sensors (TGS and MP-4) in the model for the estimation of the gas concentration? In other words, what was the input of the neural network, which was supposed to be the output voltage of the sensors?

- As before, remove unnecessary decimal digits

Overall, I found it quite hard to understand the flow of the paper due to the low quality of the English language. Several sentences should be revised and basic punctuation rules should be followed. I kindly ask the authors to read again all the paper and extensively improve the quality of the English language that they use. Some examples:

- Remove comma from: The gas sensor array equivalent to primary olfactory neurons, is composed of multiple independent gas sensors.

- Revise sentence, it is just too complex (too many subjects and verbs, ...) for the simple meaning that it has: The output voltage values of the sensors and gas concentration values in the environment were used to train a constructed neural network to establish a non-linear input and output relationship to realize the calibration of the sensor array.

Round 2

Reviewer 1 Report

Recommend publishing without any further revision

Author Response

thank you